# Identification of Causal Genes and Potential Drug Targets for Restless Legs Syndrome: A Comprehensive Mendelian Randomization Study

**DOI:** 10.3390/ph17121626

**Published:** 2024-12-04

**Authors:** Ruiyi Qian, Xue Zhao, Dongbin Lyu, Qingqing Xu, Kai Yuan, Xin Luo, Wanying Wang, Yang Wang, Yutong Liu, Yu Cheng, Yingting Tan, Fan Mou, Chengmei Yuan, Shunying Yu

**Affiliations:** 1Shanghai Mental Health Center, Shanghai Jiao Tong University School of Medicine, Shanghai 200030, China; ruiyiqian99@163.com (R.Q.); zhaoxueyujian@sjtu.edu.cn (X.Z.); shuiysuper@foxmail.com (D.L.); hualixqq@163.com (Q.X.); xinluoflora@163.com (X.L.); wangwy_w@163.com (W.W.); wangyang_sjtu@163.com (Y.W.); yutong_liu1202@163.com (Y.L.); chengyucq@163.com (Y.C.); tanyt91@163.com (Y.T.); moufan2021@163.com (F.M.); 2State Key Laboratory of Digestive Disease, Li Ka Shing Institute of Health Sciences, Institute of Digestive Disease, Department of Medicine and Therapeutics, The Chinese University of Hong Kong, Hong Kong SAR 999077, China; yuankai2017@sibcb.ac.cn

**Keywords:** restless legs syndrome, drug discovery, mendelian randomization, proteomics

## Abstract

**Background:** Restless legs syndrome (RLS) is a common sensorimotor sleep disorder that affects sleep quality of life. Much effort has been made to make progress in RLS pharmacotherapy; however, patients with RLS still report poor long-term symptom control. **Methods:** Comprehensive Mendelian randomization (MR) was performed to search for potential causal genes and drug targets using the cis-pQTL and RLS GWAS data. Robustness was validated using the summary-based Mendelian randomization (SMR) method and co-localization analysis. Further evidence of pleiotropy of the target genes and their potential side effects was provided by phenome-wide MR analysis (MR-PheWAS). Finally, molecular docking simulations were conducted on drug candidates corresponding to these targets, which revealed promising binding affinities and interaction patterns and underscored the druggable potential of the target gene. All of the analyses above were conducted in the context of *Homo sapiens*. **Results:** *MAN1A2* showed a statistically significant result in the MR analysis, which was validated through SMR and co-localization analysis. The MR-PheWAS showed a low probability of pleiotropy and prospective side effects. Molecular docking was used to visualize the binding structure and fine affinity for *MAN1A2* and the drugs predicted by DSigDB. **Conclusions:** Our study provides comprehensive evidence supporting *MAN1A2* as a promising causal gene and therapeutic target for RLS, offering insights into the underlying molecular mechanisms and paving the way for future drug development efforts.

## 1. Introduction

Restless legs syndrome (RLS) is a common sensorimotor sleep disorder [1,2,3], with a presumed prevalence of 5–8% in the general population [4,5]. The clinical manifestations of RLS include a constant urge to move while at rest, particularly at night or in the evening, accompanied by discomforting feelings [6]. This urge tends to alleviate upon movement and exacerbates during rest [6,7]. In addition to causing difficulty falling asleep, RLS often affects patients’ mood, cognition, energy, and ability to perform daily activities [7,8]. In severe cases, these symptoms have been linked to an increased risk of depression, self-harm, and suicide [8], exerting a considerable adverse impact on sleep quality, overall well-being, and health status [6].

Until now, the pathogenesis of RLS has been still unclear. Recent advances in understanding its pathophysiology have underscored the involvement of neurotransmitter dysfunction (predominantly dopamine [9], glutamate [10], and adenosine [11]), iron deficiency [12,13], and other as-yet-unidentified contributing mechanisms. While dopaminergic therapy is the primary pharmacotherapy for RLS [14,15], its prolonged usage may lead to augmentation, namely, the worsening of symptoms [16]. Although novel therapeutic alternatives, such as pregabalin [17], gabapentin enacarbil [15,17], oxycodone–naloxone [16,17], and iron supplementation [18], have been introduced in recent clinical studies, many patients still report insufficient long-term symptom control or unbearable adverse drug events [1]. Hence, it is imperative for researchers to develop more efficacious and safer pharmaceutical interventions for RLS.

Additionally, RLS prominently features a genetic underpinning according to a large-scale family study showing vertical transmission in 90% of all RLS families included [19,20]. A large genome-wide association study (GWAS) identified 22 genetic loci associated with RLS and highlighted genetic correlations between RLS and neuropsychiatric traits [19]. Another recent large-scale meta-analysis by Schormair et al. expanded the number of known RLS risk loci to 164 and highlighted neurodevelopmental pathways and potential drug targets, with machine learning models showing high predictive accuracy for RLS risk [21]. However, the causal relationship between the targets and RLS has not been studied and discussed.

The integration of genetic insights into drug development stands out as a potent strategy to enhance efficacy [22], given that therapies with genetic evidence demonstrate a greater likelihood of success in clinical trials. Meanwhile, proteins encoded by genes that serve as drug targets often harbor specific binding sites or domains amenable to interactions with certain pharmaceutical agents [23]. Recently, the UK Biobank released its plasma proteomic project (UKB-PPP), presenting comprehensive protein quantitative trait locus (pQTL) mapping of 2923 proteins and identifying 14,287 primary genetic associations [24]. These endeavors hold promise for discovering prospective biomarkers and risk/protective factors associated with RLS, in addition to enabling the assessment of the causative effects of plasma proteins on the disease.

Mendelian randomization (MR) is a statistical methodology that leverages genetic variability to elucidate causal associations between exposure and outcomes [25]. Such genetic variants are typically impervious to confounding variables and not affected by postnatal behavior, psychology, or socioeconomic factors [26].

In this study, we employed a comprehensive approach integrating MR, summary-based MR (SMR), gene co-localization analysis, phenome-wide MR analysis (MR-PheWAS), gene enrichment analysis, gene prediction, and molecular docking to provide insights into potential drug targets and factors for RLS based on genetic evidence within the context of *Homo sapiens*.

## 2. Results

### 2.1. MR Analysis

We employed GWAS data for RLS from the meta-analysis conducted by Didriksen et al. [19]. and UKB-PPP pQTL data [24] as our discovery cohorts. Through MR analysis, *MAN1A2* was identified to mitigate the risk of RLS and exhibited a significant MR result (OR = 0.68 [0.60,0.78], *p*_FDR = 1.01 × 10^−4^). Our preliminary analysis did not reveal any significant heterogeneity or pleiotropy associated with *MAN1A2*, suggesting a robust and consistent association between this protein and RLS risk across our study cohorts (Figure 1a). Moreover, *MAN1A2* showed a significantly lower *p*-value than other plasma proteins (Figure 1b). According to the inverse variance weighted (IVW) and Wald ratio methods, another 89 proteins may also have a causal relationship with RLS, but this relationship does not appear to be significant after false discovery rate (FDR) correction (Appendix A).

Detailed MR results are presented in Figure 1. The instrumental variables for the proteins included in the analysis are listed in Appendix A. Detailed outcomes of heterogeneity, pleiotropy, leave-one-out, and single nucleotide polymorphisms (SNPs) in the MR analysis are displayed in Appendix A.

Both GWAS datasets were of European ancestry and were from different cohorts so that racial differences and sample overlap could be avoided.

### 2.2. SMR Analysis

We further validated the relationship between the cis-pQTL of *MAN1A2* and RLS (*p* = 3.27 × 10^−5^, *p*_HEIDI = 0.065).

In addition, we used cis-eQTL data from GTEx, eQTLgen, and PsychENCODE to further validate the relationship and determine tissue specificity. The cis-eQTL data of *MAN1A2* were extracted from only 11 of 52 tissues and were significantly expressed in five tissues, including cultured fibroblasts, esophageal gastroesophageal junction, lung, testis, and thyroid (Table 1).

The details of cis-eQTL data used in this study and the SMR results can be found in Appendix A.

### 2.3. Co-Localization Analysis

Previous studies have shown that false-positive MR outcomes can arise from a locus with a close linkage disequilibrium (LD) between two SNPs and two causal SNPs underlying exposure/outcome associations [27]. In cases where there is an unmistakable link between exposure and outcome, co-localization analysis can be used to investigate whether exposure and outcome share the same causal SNP [28]. It has been found that therapeutic targets are more likely to be approved for proteins that have passed both MR and co-localization tests [23].

To ascertain whether there may be common causative genetic variations linked to RLS and pQTL, we performed co-localization analysis on *MAN1A2*. Specifically, the posterior probabilities (PP) associated with each hypothesis were PP.H0 = 8.77 × 10^−147^, PP.H1 = 2.17 × 10^−42^, PP.H2 = 4.38 × 10^−147^, PP.H3 = 9.90 × 10^−3^, and PP.H4 = 0.968. These results indicate a high probability (PP.H4 = 0.968) of shared causal genetic variation underlying both RLS and pQTL for *MAN1A2*.

### 2.4. Phenome-Wide Association Analysis (PheWAS)

We used 2407 phenotypes from the FinnGen [29] to run MR-PheWAS at the gene level in order to further evaluate if the putative drug target genes discovered would have other potential pleiotropy that failed to be detected by the MR-Egger intercept test. The relationship between the pQTL of *MAN1A2* and particular diseases or features can be inferred from MR-PheWAS results. Thyrotoxicosis, psychotic depression, tympanosclerosis, vascular dementia, and esophageal ulcers showed the highest genetic correlations with *MAN1A2* (Figure 2). However, none of these were significantly related (*p*_FDR < 0.05), indicating that the likelihood of side effects from drugs acting on the target and the presence of horizontal pleiotropy was likely to be low, further demonstrating the validity of the study’s findings. The details of the MR-PheWAS outcome can be found in Appendix A.

### 2.5. Pathway and Functional Analysis

In the Gene Ontology (GO) enrichment analysis, genes were enriched for biological terms [30], whereas in the Kyoto Encyclopedia of Genes and Genomes (KEGG) enrichment analysis [31], they were enriched for functional pathways. As shown in Appendix A, the most significant pathways in the BP category were certain protein processes and the respiratory system. In the MF class, *MAN1A2* participates in glycosyl and mannosidase activity. Regarding KEGG enrichment, *MAN1A2* was also shown to be related to glycans and protein processing.

### 2.6. Candidate Drug Prediction

Using the DSigDB database, 25 chemical compounds were predicted to be potentially effective intervention drugs (Table 2). Antimony potassium tartrate (CTD 00005424) had the lowest *p*-value.

### 2.7. Molecular Docking

Molecular docking was utilized to evaluate the binding affinity between drug candidates and their targets, with the aim of identifying druggable targets. Autodock Vina (version 1.5.7) was used to obtain the binding sites and interactions of all 25 drug candidates with *MAN1A2* and to generate the binding energy for each interaction (Table 2). Every pharmaceutical candidate forms observable hydrogen bonds and significant electrostatic interactions with its protein target, with the exception of antimony since it is an atom. The compound sanguinarine (MCF7 DOWN) demonstrated the most favorable binding energy of −9.5 kcal/mol, suggesting a highly persistent binding interaction (Figure 3).

## 3. Discussion

In this study, we identified *MAN1A2* as a potential therapeutic target for RLS based on multiple MR methods complemented by rigorous analyses. The robustness of the findings was further validated and supported by the SMR method and co-localization analysis. Pleiotropy and prospective side effects were unlikely according to MR-PheWAS analysis. Enrichment analysis revealed the biological functions of *MAN1A2*. Finally, drug candidates were predicted and subjected to molecular docking simulations, which revealed promising binding affinities and interaction patterns and underscored the druggable potential of the target gene.

Our study identified *MAN1A2* as a novel gene associated with RLS, a finding that has not been reported in previous genetic studies. While prior GWAS studies have identified multiple loci linked to iron metabolism, neurotransmitter regulation, and neuronal signaling in the context of RLS [19,21,32], *MAN1A2* represents a distinct pathway that broadens our understanding of the disease’s etiology.

*MAN1A2* is located on chromosome 1p13, which belongs to the mannosidase alpha class 1 gene group and encodes mannosyl-oligosaccharide 1,2-alpha-mannosidase IB, located at the Golgi apparatus membrane subcellularly [33]. Alpha-mannosidases function at various stages of N-glycan maturation in mammalian cells [33,34]. After the transfer of precursor Glc3Man9GlcNAc2 to polypeptides and removal of glucose residues from the endoplasmic reticulum, Man5GlcNAc2 is formed through joint action in the endoplasmic reticulum, and the Golgi apparatus cleaves up to four α1,2-linked mannose residues [33].

A previous clinical study discovered that 24 N-glycan structures exhibited notable disparity and disproportionality between the control and RLS groups [2]. This suggests that N-glycans have the potential to serve as glycan biomarkers for distinguishing RLS from healthy individuals and that *MAN1A2* could be a promising target for drug intervention in RLS treatment by adjusting N-glycan levels.

In addition, RLS and N-glycans are closely associated with immunology and inflammation. Prior results from N-glycan processing gene arrays have revealed *MAN1A2* as a target for downregulation by TNF-α [35]. TNF-α can modulate the pattern of N-glycosylation in synoviocytes [36] and mediate interconversion between high-mannose and hybrid N-glycans in epithelial cells [37]. As for RLS, Leonard B. Weinstock et al. [38]. reviewed all conditions that were reported to have an association with RLS, of which 89% are related to inflammatory and/or immune changes. TNF-α is an elevated cytokine indicator and mediator under pertinent conditions [38]. Therefore, it is presumed that *MAN1A2* may affect RLS occurrence through the mediation of N-glycans and immune/inflammatory mechanisms and that increased TNF-α is a pivotal inflammatory change in the mechanism of RLS.

Beyond its role in RLS, *MAN1A2* has been implicated in broader biological processes and diseases, especially cancers [39,40,41,42], where it serves as a potential biomarker. Shared inflammatory pathways, such as their regulation by TNF-α [35], may connect *MAN1A2* to immune-related mechanisms common to both cancers and RLS. Elevated TNF-α levels, a hallmark of inflammation, have been observed in RLS and other conditions, suggesting that *MAN1A2* may act at the intersection of glycosylation and inflammation. To address its specific role in RLS, future research should examine its expression and integrate multi-omics approaches to clarify disease-specific versus systemic roles. These studies will help refine its therapeutic potential and mitigate the risk of off-target effects in other inflammatory or oncologic conditions.

Among the drugs predicted by DsigDB, with *MAN1A2* as the drug target, there are eight cardiotonic drugs (ouabain, strophanthidin, sanguinarine, digoxin, digoxigenin, digitoxigenin, lanatoside C, and proscillaridin), with their pharmacological mechanisms mainly involving blocking Na^+^-K^+^-ATPase (NKA). Markina et al. [43] recently reported the effects of altered NKA function on dopamine signaling and metabolism. In addition, cycloheximide and lycorine are drugs used to treat cardiovascular diseases, revealing the possible role of cardiovascular diseases and their therapeutic targets in the treatment of RLS. Large population-based studies have found cardiovascular diseases to be risk factors for RLS [44], and patients with RLS show a significant increase in the risk of cardiovascular disease [45,46]. The observed correlations may be attributed to several shared mechanisms between the two diseases, such as increased sympathetic activity, oxidative stress, metabolic factors, and inflammation [47]. Thus, RLS is presumed to have drug targets similar to those in cardiovascular diseases.

In addition, ouabain, anisomycin, and apigenin act on the dopaminergic pathway and can elevate dopamine levels. Ouabain is proven to increase dopamine levels in the striatum and cerebrospinal fluid of mice [48,49]. Local infusion of anisomycin can cause a rapid dopamine level increase at the site [50,51]. Apigenin can also raise dopamine levels in mice and drosophila [52,53]. The role of the dopaminergic system in the development of RLS symptoms is confirmed by the rapid improvement observed when low doses of most dopaminergic agents are administered and when the use of dopamine antagonists leads to exacerbation [54,55]. Levodopa, a precursor of dopamine, was used as the primary treatment for RLS. However, due to their short half-life and potential for symptom augmentation, dopamine agonists with longer half-lives have become the preferred first-line agents for RLS treatment [56]. Nonetheless, long-term treatment with dopamine agonists may lead to diminished efficacy and the emergence of side effects [57]. Therefore, other drugs that act on the dopaminergic pathway with a longer half-life, drugs that locally increase dopamine concentrations, and dopamine analogs as possible new drugs are still promising.

The exploration of novel medications for RLS can be approached by examining these substances and their mechanisms. Modifications can potentially transform the relevant medicines into novel treatments for RLS.

This study provides strong evidence to suggest that *MAN1A2* is an effective drug target for RLS, based on a comprehensive evaluation of drug-binding properties. Using data from the largest publicly accessible RLS GWAS, our study was the first to use MR to identify RLS drug targets. The results were further validated by SMR and co-localization methods to reduce the possibility of false positives. The functional characteristics of *MAN1A2* were illustrated using enrichment analysis. Molecular docking demonstrated high binding activity of *MAN1A2*, further validating its therapeutic potential.

Study limitations include the fact that results obtained from MR studies may not accurately reflect the magnitudes of effect sizes found in real-world clinical situations and that they may not entirely predict the impact of a medicine, even after employing several validation methods and techniques [58]. Therefore, it is essential to perform additional empirical verification and meticulous clinical studies to ensure the therapeutic potential of the target and evaluate the safety and efficacy of these drugs.

Another constraint is derived from the diversity of the study sample. While eQTLs analysis includes individuals of non-European heritage, the GWAS data for cis-pQTL and RLS are exclusively limited to European individuals. As a result, the study’s capacity to be applied to other populations is restricted because of its major focus on people of European heritage. Additional studies and validations are necessary to determine the broader relevance of these findings in people of different ethnicities.

Furthermore, cis-pQTLs and their relationship with RLS were the primary focus of this study. Despite rigorous endeavors to reduce bias, MR analysis is still susceptible to unmeasured factors or pleiotropy as well as other factors that contribute to the intricate pathogenesis of the disease, which could impact the results. Thus, the underlying mechanisms of RLS should be better understood by including omics data and environmental factors in future studies.

Finally, the quality of the protein structures and ligands greatly affects the accuracy of molecular docking analysis. Drug targets and potential drugs have been identified using this approach, but their effectiveness in clinical situations is not guaranteed. Further experimental validation and clinical trials are necessary to confirm the safety and effectiveness of these selected targets.

## 4. Methods

### 4.1. Data Source

The genetic data for RLS stem from a large-scale GWAS involving 480,982 individuals of European ancestry, comprising 10,257 RLS cases and 470,725 controls [19]. Clinical diagnosis or questionnaire-based assessments, specifically employing the International Restless Legs Syndrome Study Group (IRLSSG) diagnostic criteria or the Cambridge-Hopkins Restless Legs Syndrome questionnaire (CH-RLSq), were used to ascertain RLS status among the participants. Summary data were obtained from deCODE.

Proteomic data originate from the UK Biobank Pharma Proteomics Project (UKB-PPP) [24], wherein blood plasma samples were obtained from 54,219 UKB participants. Sun et al. [24]. employed the antibody-based Olink Explore 3072 PEA to perform proteomic profiling, measuring 2941 protein analytes and capturing 2923 unique proteins. For detailed information on the sources of the summary data, please refer to the “Data Availability” section.

Single nucleotide polymorphisms (SNPs) associated with each plasma protein were retained if they exhibited a minor allele frequency of at least 1% and achieved genome-wide significance (*p* < 5 × 10^−8^). Moreover, SNPs demonstrating high linkage disequilibrium (LD) with each other (with an LD R^2^ value exceeding 0.1 in the 1000 Genomes Project from the European population) were deemed redundant and excluded from subsequent analyses.

The other data sources used in the extended analysis are described in detail in their respective sections.

### 4.2. MR Analysis

We used MR to examine the connection between plasma proteins and RLS and strictly adhered to the recommendations provided by the Strengthening the Reporting of Observational Studies in Epidemiology–Mendelian randomization (“STROBE-MR”) framework [59].

Following three fundamental assumptions, we applied a series of filtering criteria to select appropriate genetic instruments: ① a significance threshold of *p* < 5 × 10^−8^; ② ensuring sufficient instrument strength, as indicated by an F-statistic > 10; ③ imposing a minimum physical distance of 1000 kb between any two genetic variants; and ④ maintaining an LD threshold of r^2^ < 0.1 between any two genes (based on the 1000 Genomes European reference panel) [60].

Subsequent analysis was conducted using Wald ratios for proteins instrumented with a single SNP and the inverse variance weighted (IVW) method for proteins instrumented with more than one SNP. False discovery rate (FDR) correction served as an implementation to mitigate the risk of false discoveries in multiple comparisons, with a threshold of *p*_FDR < 0.05 deemed significant.

Cochran’s Q statistic was employed to assess the heterogeneity of causal effects, and MR-Egger’s intercept term for horizontal pleiotropy was used in the external validation. In these tests, *p*-values below 0.05 usually suggest the existence of heterogeneity or pleiotropy [61]. The Steiger test was applied to address potential issues of reverse causality [62].

### 4.3. SMR Analysis

Summary-based Mendelian randomization (SMR) and heterogeneity in dependent instruments (HEIDI) provide a statistical framework for assessing whether gene expression mediates the effect size of an SNP on a phenotype through integration with xQTL data [63]. The statistic *p*_FDR < 0.05 was considered significant, and *p*_HEIDI > 0.05 was considered independent of LD. To validate our findings and address tissue specificity, we conducted SMR analysis for RLS using expression quantitative trait locus (eQTL) datasets from the Genotype-Tissue Expression Project (GTEx), the blood eQTL dataset from eQTLgen, and cerebrospinal fluid eQTL from PsychENCODE.

### 4.4. Co-Localization Analysis

Co-localization analysis represents a pivotal methodological step aimed at reinforcing the findings of genetic investigations by identifying evidence of shared genetic variation associated with both exposure and outcome variables. From this analysis, it can be concluded that genetic variation genuinely contributes to the outcome and does not simply reflect linkage disequilibrium (LD) or other confounding factors [28].

Bayesian co-localization analysis was conformed to evaluate the following five mutually exclusive hypotheses: H0, indicating no association between a genetic variant and any trait; H1, suggesting an association solely with one trait; H2, indicating association solely with another trait; H3, signifying an association with both traits but through distinct causal variants; and H4, denoting an association with both traits and sharing the same causal variant [28]. Posterior probabilities (PPs) were calculated for each hypothesis, and the presence of co-localization evidence for a protein was determined based on PP.H4 > 0.8 threshold.

### 4.5. MR-PheWAS

To comprehensively assess possible side effects and horizontal pleiotropy of the potential drug target not captured by the MR-Egger intercept test, a phenome-wide MR study (MR-PheWAS) was performed. The positive pQTLs obtained in the previous step were combined with all 2408 phenotypes in FinGenn R10 [29] for MR-PheWAS. The filtering criteria to select appropriate genetic instruments for all phenotypes were the same as in the previous MR analysis, and FDR correction was applied to avoid insufficient evidence.

### 4.6. Pathway and Functional Analysis

To investigate the functional attributes and biological relevance of the identified prospective therapeutic target genes, we conducted Gene Ontology (GO) enrichment analysis [30] and Kyoto Encyclopedia of Genes and Genomes (KEGG) pathway enrichment analysis [31].

GO analysis encompasses three distinct categories: biological processes (BP), molecular functions (MF), and cellular components (CCs). KEGG pathway enrichment analysis provides valuable information about the metabolic and signaling pathways in which the target genes participate, offering a broader understanding of their roles within biological systems.

### 4.7. Statistical Analysis

All analyses above were performed using the TwoSampleMR (version 0.5.8), MendelianRandomization (version 0.8.0), BioConductor (version 3.18), clusterProfiler (version 4.10.0), coloc (version 5.2.3), and locuscomparer (version 1.0.0), etc., in R Software 4.3.2 (https://www.R-project.org, accessed on 1 December 2023).

### 4.8. Candidate Drug Prediction

Assessing protein–drug interactions is crucial for determining the feasibility of utilizing target genes as potential drug targets. In this study, we used the Drug Signatures Database (http://dsigdb.tanlab.org/DSigDBv1.0/, accessed on 25 January 2024) [64]. Specifically, DSigDB comprises a substantial repository of 22,527 gene sets and 17,389 distinct compounds spanning 19,531 genes. This extensive database serves as a valuable resource for establishing connections between medications, other chemical entities, and their respective target genes.

To evaluate their therapeutic potential, drug candidates were predicted by uploading the identified target genes to the DSigDB. Using DSigDB, we hope to gain insights into the medicinal activity of target genes, thereby facilitating the identification of promising drug candidates.

### 4.9. Molecular Docking

Through molecular docking at the atomic level, researchers can gain a deeper understanding of how drug candidates affect target genes and assess their drug abilities. This allows for the prediction of interactions between ligands (drug candidates) and target proteins based on their binding affinity. The identification of ligands with high binding affinities and advantageous interactions can aid in optimizing therapeutic candidates and selecting pharmacological targets for further testing.

Drug candidates were docked with their respective target proteins using AutoDock Vina (version 1.5.7) [65] (http://autodock.scripps.edu/, accessed on 28 January 2024).

Drug structure data were sourced from the PubChem Compound Database [66] (https://pubchem.ncbi.nlm.nih.gov/, accessed on 28 January 2024) using the ChemSpider Database [67] (https://www.chemspider.com/, accessed on 28 January 2024) if necessary.

Protein structural data were obtained from the AlphaFold Protein Structure Database [68] (https://alphafold.ebi.ac.uk/, accessed on 28 January 2024). A pocket size of 50 × 50 × 50 was used to construct the docking pocket to ensure the comprehensive coverage of the potential binding sites. The entire molecular docking process was visualized in the model using Autodock Vina 1.5.7 and Pymol 3.0 [69], facilitating the interpretation of binding interactions and the identification of ligands exhibiting high binding affinity and favorable interaction patterns.

A flowchart depicting the methodology used in this study is shown in Figure 4.

### 4.10. Ethics Approval

According to the original RLS GWAS article, the deCODE dataset was approved by the National Bioethics Committee of Iceland. The DBDS dataset was approved by the Scientific Ethical Committee of Central Denmark (M-20090237) and by the Danish Data Protection Agency (30-0444). GWAS studies in DBDS were approved by the National Ethical Committee (NVK-1700407). The INTERVAL dataset was approved by the National Research Ethics Service Committee East of England—Cambridge East (Research Ethics Committee (REC: 11/EE/0538). The Emory dataset was approved by an institutional review board at Emory University, Atlanta, Georgia, US (HIC ID 133-98). The Donor InSight-III dataset was approved by the Medical Ethical Committee of the Academic Medical Center (AMC) in the Netherlands. Sanquin’s Ethical Advisory Board approved DIS-III, and all participants gave their written informed consent. UK Biobank is approved by the North West Multi-center Research Ethics Committee, the Patient Information Advisory Group, the National Information Governance Board for Health and Social Care, and the Community Health Index Advisory Group. UK Biobank also holds a Human Tissue Authority license.

The ethics approval of GTEx, eQTLgen, and psychENCODE can be found at https://www.nature.com/articles/ng.2653#ethics (accessed on 28 September 2024), https://www.nature.com/articles/s41588-021-00913-z#ethics (accessed on 28 September 2024) and https://www.nature.com/articles/nn.4156#ethics (accessed on 28 September 2024).

The Finngen research project was evaluated by the Coordinating Ethics Committee of the Helsinki and Uusimaa Hospital District, and it is based on samples from Finnish biobanks and data from national health registers. The study applies the permission to utilize the register data for research purposes from national authorities. The research project complies with existing legislation (in particular, the Biobank Law and the Personal Data Act) and will conform to any new laws. The EU Data Protection Regulation that came into force in May 2018 has been taken into account when planning the project.

Given that all data are publicly accessible and fully anonymized, the study did not require a new ethics committee or institutional review board approval. This is in accordance with standard guidelines for secondary data analysis of public datasets.

## 5. Conclusions

This comprehensive Mendelian randomization study provides robust evidence supporting *MAN1A2* as a protective plasma protein for RLS, with various analyses enhancing the robustness. Drug prediction and molecular docking were implemented to validate the medicinal value of *MAN1A2* as a therapeutic target. These insights not only deepen our understanding of RLS pathophysiology but also open new avenues for therapeutic intervention. Future studies are warranted to validate these findings and explore their clinical applications in managing and preventing RLS.

## Figures and Tables

**Figure 1 pharmaceuticals-17-01626-f001:**
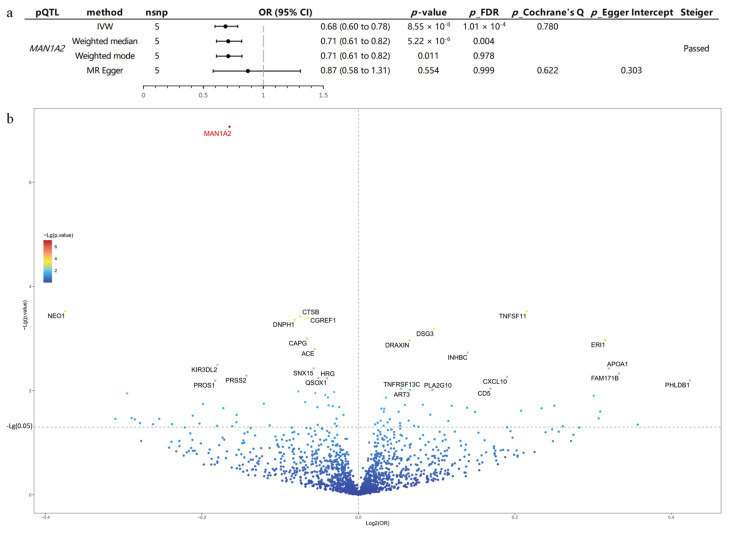
(**a**) Forest plot displaying the positive findings from MR analysis. *MAN1A2* exhibits a significant causal association (*p*_FDR < 0.05, *p*_Cochrane’s Q > 0.05 and *p*_Egger Intercept > 0.05) with RLS outcome. According to IVW, *MAN1A2* was identified to mitigate the risk of RLS and exhibited a significant MR result (OR = 0.68 [0.60,0.78], FDR = 1.01 × 10^−4^). (**b**) Volcano plots of the MR results between plasma proteins and RLS. The *x*-axis represents the log2 odds ratio (log2(OR)) of the causal effect of each protein, while the *y*-axis shows the negative logarithm of the *p*-value (−Lg(*p*-value)) for each association. The dashed horizontal line indicates the significance threshold (*p* = 0.05). The color gradient represents the level of significance, with warmer colors indicating stronger associations. The *p*-value of *MAN1A2* is significantly lower than other proteins. All plasma proteins with *p*-value < 0.01, *p*_Cochrane’s Q > 0.05, and *p*_Egger Intercept > 0.05 were labeled; however, only *MAN1A2* still showed significant association with RLS after FDR correction. FDR, False discovery rate; *p*_FDR, *p*-value after FDR correction; *p*_Cochrane’s Q, *p*-value from Cochrane’s Q test; *p*_Egger Intercept, *p*-value from MR Egger analysis; OR, odds ratio.

**Figure 2 pharmaceuticals-17-01626-f002:**
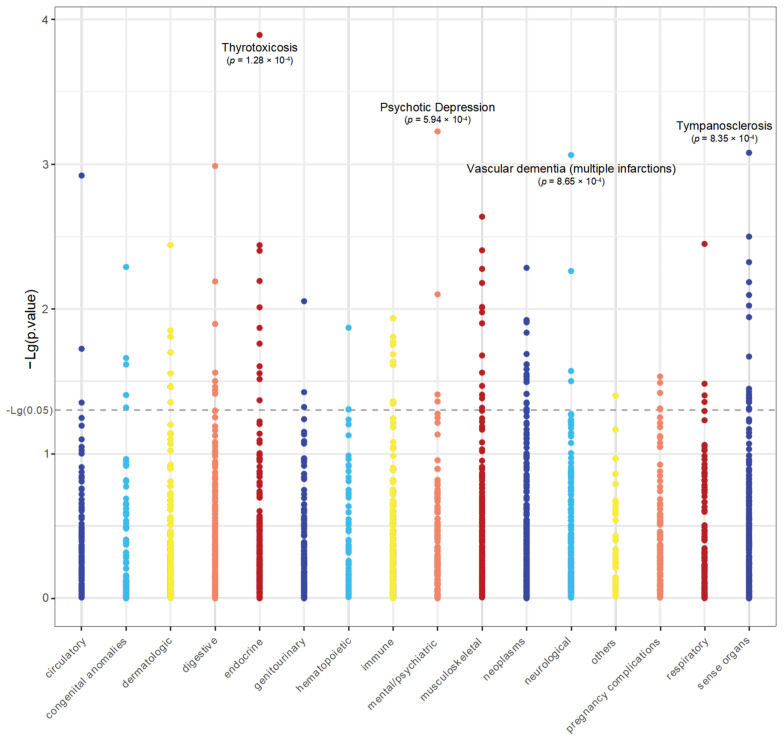
Manhattan plot displaying MR-PheWAS findings. This plot illustrates the associations between *MAN1A2* (from UKB-PPP) and 2407 phenotypes (from FinnGen) grouped into major biological categories. The *x*-axis represents phenotype categories, while the *y*-axis shows the negative logarithm of the *p*-values (−Lg(*p*-value)) for each association. The dashed horizontal line indicates the significance threshold (−log10(0.05)). *MAN1A2* was not significantly related to the phenotypes after FDR correction. Four diseases with *p*-value < 0.0001 were labeled, but the relationship does not appear to be significant after FDR correction.

**Figure 3 pharmaceuticals-17-01626-f003:**
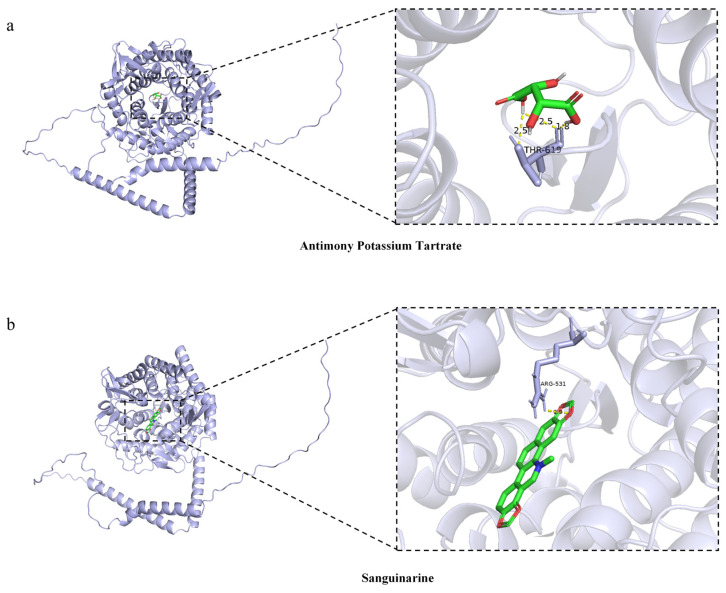
Molecular docking results of *MAN1A2* and predicted drugs. (**a**) the interaction between *MAN1A2* and antimony potassium tartrate; (**b**) the interaction between *MAN1A2* and sanguinarine. The blue ribbons represent the three-dimensional structure of the *MAN1A2* protein. The green sticks represent the predicted drugs. The red portions of the ligands represent oxygen-containing groups, the blue portions represent nitrogen-containing groups, and the white portions represent hydrogen atoms. The yellow dotted lines indicate potential hydrogen bonds between *MAN1A2* and the corresponding drugs. The labeled residues (e.g., THR-619 and ARG-531) are the key amino acid residues of MAN1A2 that interact with the drugs, and the numbers (e.g., 2.5, 1.8) represent the hydrogen bond lengths in angstroms.

**Figure 4 pharmaceuticals-17-01626-f004:**
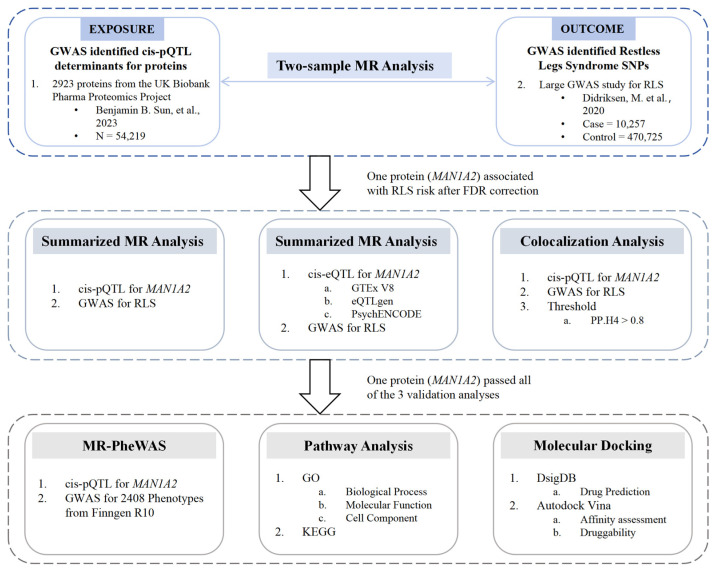
Flow chart of the study design. The exposure includes cis-pQTL determinants for 2923 proteins identified in the UKBPPP (Benjamin B. Sun et al., 2023) [24]. The outcome is the GWAS data for RLS (Didriksen M et al., 2020) [19]. GWAS, genome-wide association study; MR, Mendelian randomization; pQTL, protein quantitative trait loci; eQTL, expression quantitative trait loci; RLS, restless legs syndrome; PheWAS, phenome-wide association study; GO, Gene Ontology; KEGG, Kyoto Encyclopedia of Genes and Genomes.

**Table 1 pharmaceuticals-17-01626-t001:** Validation and tissue specificity findings from SMR analysis.

Data Source	Tissue	*p*_SMR	*p*_HEIDI
GTEx	Adipose Visceral Omentum	0.174	0.139
GTEx	Artery Tibial	0.247	0.108
GTEx	Cells Cultured fibroblasts *	0.013	0.073
GTEx	Esophagus Gastroesophageal Junction *	8.27 × 10^−4^	0.914
GTEx	Lung *	5.11 × 10^−4^	0.695
GTEx	Muscle Skeletal	0.238	0.013
GTEx	Nerve Tibial	0.054	0.05
GTEx	Skin Not Sun Exposed Suprapubic	0.495	0.045
GTEx	Skin Sun Exposed Lower leg	0.495	0.036
GTEx	Testis *	0.023	0.486
GTEx	Thyroid *	0.017	0.088
eQTLgen	Blood	0.350	/
PsychENCODE	Cerebrospinal Fluid	0.892	0.493

The cis-eQTL data of *MAN1A2* were extracted from only 11 of 52 tissues and were significantly expressed in the above five tissues. SMR, summary-based Mendelian randomization; HEIDI, heterogeneity in dependent instruments; * *p*_SMR < 0.05 and *p*_HEIDI > 0.05. *p*_SMR, *p*-value from SMR analysis; *p*_HEIDI, *p*-value from HEIDI test.

**Table 2 pharmaceuticals-17-01626-t002:** Candidate drugs predicted by DSigDB and their affinity predicted by Autodock Vina.

Drug Names	Collection	*p*-Value	*p*_FDR	Affinity (kcal/mol)
Antimony	CTD 00005424	0.004	0.044	/
Antimony Potassium Tartrate	CTD 00005425	0.004	0.044	−5.4
Phenol	CTD 00007305	0.005	0.044	−4.8
Cycloheximide	MCF7 DOWN	0.007	0.044	−7.9
Ouabain	PC3 DOWN	0.007	0.044	−7.1
Strophanthidin	PC3 DOWN	0.008	0.044	−7.4
Midecamycin	HL60 UP	0.009	0.044	−6.9
Cephaeline	MCF7 DOWN	0.010	0.044	−8.8
Emetine	MCF7 DOWN	0.012	0.044	−8.2
Niclosamide	HL60 DOWN	0.013	0.044	−8.2
Verteporfin	HL60 DOWN	0.014	0.044	−5.9
Lycorine	PC3 DOWN	0.015	0.044	−8.6
Anisomycin	MCF7 DOWN	0.015	0.044	−7.0
Sanguinarine	MCF7 DOWN	0.019	0.044	−9.5
Apigenin	HL60 DOWN	0.020	0.044	−8.3
Thapsigargin	MCF7 UP	0.020	0.044	−7.2
Digoxin	HL60 DOWN	0.022	0.044	−8.5
Helveticoside	HL60 DOWN	0.027	0.044	−8.8
Primaquine	PC3 DOWN	0.027	0.044	−6.9
Lanatoside C	HL60 DOWN	0.028	0.044	−5.7
Proscillaridin	HL60 DOWN	0.031	0.045	−8.1
Digoxigenin	HL60 DOWN	0.033	0.045	−7.6
Trichostatin A	HL60 UP	0.033	0.045	−7.6
Digitoxigenin	PC3 DOWN	0.035	0.045	−7.4
Vorinostat	HL60 UP	0.036	0.045	−7.6

According to DSigDB database, 25 chemical compounds were predicted to be potentially effective intervention drugs for restless legs syndrome. Antimony potassium tartrate (CTD 00005424) had the lowest *p*-value, and sanguinarine (MCF7 DOWN) demonstrated the most favorable binding energy. Antimony is an atom, so the affinity is not available through molecular docking. FDR, false discovery rate; *p*_FDR, *p*-value after FDR correction.

## Data Availability

We thank the UK Biobank, deCODE, Yang Lab, and FinnGen for sharing data and tools. The SMR tool was downloaded from “https://yanglab.westlake.edu.cn/software/smr” (accessed on 10 December 2023). GWAS data for RLS were downloaded from “https://www.decode.com/summarydata/” (accessed on 6 December 2023). We really appreciate Sun et al. [24] for sharing UKB-PPP GWAS summary data on “https://metabolomips.org/ukbbpgwas/” (accessed on 6 December 2023). The raw UKB-PPP data can be obtained through an interactive portal (“http://ukb-ppp.gwas.eu”, accessed on 3 December 2023). UKB has cataloged the dataset in Category 1839, under ‘Field 30900’, described in greater detail online (“https://biobank.ndph.ox.ac.uk/showcase/label.cgi?id=1839”, accessed on 3 December 2023). GTEx v8 related data were downloaded from “https://gtexportal.org/home/” (accessed on 25 December 2023). Data from the 1000 Genomes Project were downloaded from “https://www.broadinstitute.org/” (accessed on 4 December 2023). The whole-blood eQTL data were downloaded from “https://eqtlgen.org” (accessed on 24 December 2023). The PheWAS phenotype data were obtained from “https://r10.finngen.fi” (accessed on 10 January 2024). All of the data generated during this study are included in this article and its Appendix A.

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
