# Peer review of "Identification of Causal Genes and Potential Drug Targets for Restless Legs Syndrome: A Comprehensive Mendelian Randomization Study"

_pharmaceuticals, 2024, doi:10.3390/ph17121626_

Round 1
Reviewer 1 Report
Comments and Suggestions for Authors
Identification of Causal Genes and Potential Drug Targets for Restless Legs Syndrome: a comprehensive Mendelian Randomization Study
The authors present an analysis using existing datasets of RLS, in order to identify druggable targets for the condition. The work is interesting and well-written, and needs only moderate modification in my view to be acceptable for publication. The single largest issue is the use of UK Biobank data and the importance of correctly including the UK Biobank application number.
Comments
[1] Both the Introduction and the Discussion need to set out the background to this research in terms of existing work on the topic. Several papers have been written recently on GWAS and RLS, as I am sure the authors are aware. The Introduction should provide an up-to-date summary of existing GWAS / RLS research findings, and the Discussion should explain where the authors agree, disagree or find completely new insights. Especially the work of Schormair et al (2024) is a really excellent work that the authors should be using to compare and contrast their own findings.
PMID: 39078117
PMID: 38839884
[2] The Discussion should cover the importance of MAN1A2 more generally, including appropriate references. It is interesting that MAN1A2 features as a marker of RLS but it has frequently been proposed as a marker of cancer. The authors should discuss this, and whether it derives from common inflammatory or immune mediated responses, and whether this commonality limits specificity.
-
- PMID: 31046163
[3] The Data Availability statement just says "We thank the UK Biobank" and unless I missed it, the Biobank application number is not included. Either include the Biobank application number, or state where the specific UK Biobank data were obtained.
https://www.ukbiobank.ac.uk/enable-your-research/approved-research
[4] The Introduction and Results do not explain why the researchers selected MAN1A2 as the most interesting target. Especially given that other papers (see above) have found other targets as more relevant, what was the logic / workflow behind selecting MAN1A2, and why only one target in this paper? How do I know this gene is the 'best' gene to target?
[5] The inclusion of 11 supplementary data tables is commendable and will help reproducibility. Could they be mentioned in the main manuscript, either in the text (so each table is referenced at the appropriate part of Methods or Results). Also the supplementary tables titles need to be more detailed. For example, Table S10 and S11 have identical titles, which is not helpful to the reader, as they show different things.
Author Response
Dear Reviewer,
We are grateful for your thorough and insightful comments on our manuscript, “Identification of Causal Genes and Potential Drug Targets for Restless Legs Syndrome: a comprehensive Mendelian Randomization Study”. Your feedback has greatly helped us enhance the clarity and scientific rigor of our work. Below, we address each of your comments in detail.
Comment1: Both the Introduction and the Discussion need to set out the background to this research in terms of existing work on the topic. Several papers have been written recently on GWAS and RLS, as I am sure the authors are aware. The Introduction should provide an up-to-date summary of existing GWAS / RLS research findings, and the Discussion should explain where the authors agree, disagree or find completely new insights. Especially the work of Schormair et al (2024) is a really excellent work that the authors should be using to compare and contrast their own findings.
Response1:
Thank you for suggesting a more comprehensive contextualization of our work in light of recent GWAS findings on RLS.
We have expanded the Introduction to include the following content:
[Introduction] line 55-63
Additionally, RLS prominently features a genetic underpinning according to a large-scale family study showing vertical transmission in 90% of all RLS family included[19, 20]. A large genome-wide association study (GWAS) identified 22 genetic loci associated with RLS and highlighted genetic correlations between RLS and neuropsychiatric traits[19]. Another recent large-scale meta-analysis by Schormair et al. expanded the number of known RLS risk loci to 164 and highlighted neurodevelopmental pathways and potential drug targets, with machine learning models showing high predictive accuracy for RLS risk[21]. However, the causal relationship between the targets and RLS has not been studied and discussed.
In the Discussion, we have added a comparison of our results with existing GWAS results.
[Discussion] line 204-208
Our study identified MAN1A2 as a novel gene associated with RLS, a finding that has not been reported in previous GWAS studies. While prior GWAS studies have identified multiple loci linked to iron metabolism, neurotransmitter regulation, and neuronal signaling in the context of RLS, MAN1A2 represents a distinct pathway that broadens our understanding of the disease's etiology.
Comment2: The Discussion should further explore the broader relevance of MAN1A2, including its role as a cancer marker and any shared inflammatory or immune pathways that could impact specificity for RLS.
Response2: We really appreciate this suggestion and have expanded the Discussion to address the dual role of MAN1A2 in RLS and its frequent identification as a cancer biomarker (PMID: 31046163). We discuss potential common pathways, such as inflammatory and immune-mediated mechanisms, which may explain its relevance to multiple conditions and could potentially impact specificity for RLS.
[Discussion] line 233-242
Beyond its role in RLS, MAN1A2 has been implicated in broader biological processes and diseases, especially cancers[39-42], where it serves as a potential biomarker. Shared inflammatory pathways, such as its regulation by TNF-α[35], may connect MAN1A2 to immune-related mechanisms common to both cancers and RLS. Elevated TNF-α levels, a hallmark of inflammation, have been observed in RLS and other conditions, suggesting MAN1A2 may act at the intersection of glycosylation and inflammation. To address its specific role in RLS, future researches should examine its expression and integrate multi-omics approaches to clarify disease-specific versus systemic roles. These studies will help refine its therapeutic potential and mitigate the risk of off-target effects in other inflammatory or oncologic conditions.
Comment3: The Data Availability statement just says "We thank the UK Biobank" and unless I missed it, the Biobank application number is not included. Either include the Biobank application number, or state where the specific UK Biobank data were obtained.
Response3: Thank you for pointing this out. We have updated the Data Availability section to include the UK Biobank application number and clarified the data access process.
[Data Availability] 502-505
GWAS data for UKB-PPP are available through an interactive portal (https://metabolomips.org/ukbbpgwas/). UKB has catalogued the dataset in Category 1839, under ‘Field 30900’, described in greater detail online (https://biobank.ndph.ox.ac.uk/showcase/ label.cgi?id=1839).
Comment4: The rationale for selecting MAN1A2 as the main target is not sufficiently explained in the Introduction and Results. Please clarify the selection logic and workflow, particularly given the existence of other suggested targets.
Response4: We appreciate your feedback on the need to clarify our choice of MAN1A2. We have added a volcano plot in Figure 1 to clarify that MAN1A2 has a significantly lower p value compared to other proteins. Also, we supplemented certain content in the Results section.
[2.1. MR ANALYSIS:] line 91-95
Besides, MAN1A2 showed a significantly lower p-value than other plasma proteins (Figure 1b). According to IVW and Wald Ratio, another 89 proteins may also have a causal relationship with RLS, but this relationship does not appear to be significant after FDR correction (Supplementary Table S2).
[Figure 1] line 102-114

Figure 1. (a) Forest plot displaying the positive findings from MR analysis. MAN1A2 exhibits a significant causal association (p_FDR < 0.05, p_Cochrane’s Q > 0.05 and p_Egger Intercept > 0.05) with RLS outcome. According to IVW, MAN1A2 was identified to mitigate the risk of RLS and exhibited a significant MR result (OR=0.68 [0.60,0.78], FDR =1.01E-04). (b) Volcano plots of the MR results between plasma proteins and RLS. The x-axis represents the log2 odds ratio (log2(OR)) of the causal effect of each protein, while the y-axis shows the negative logarithm of the p-value (-Lg(p-value)) for each association. The dashed horizontal line indicates the significance threshold (p = 0.05). The color gradient represents the level of significance, with warmer colors indicating stronger associations. The p-value of MAN1A2 is significantly lower than other proteins. All plasma proteins with p-value < 0.01, p_Cochrane’s Q > 0.05 and p_Egger Intercept > 0.05 were labeled, however, only MAN1A2 still showed significant association with RLS after FDR correction.
Comment5: The supplementary tables should be referenced in the main text, and titles should be more descriptive to aid reader comprehension.
Response5: Thank you for this helpful suggestion. We have added references to each supplementary table at the relevant points in the Methods section to improve accessibility for readers. Additionally, we revised the titles of the supplementary tables for greater clarity.
[2.1. MR ANALYSIS] line 92-95
According to IVW and Wald Ratio, another 89 proteins may also have a causal relationship with RLS, but this relationship does not appear to be significant after FDR correction (Supplementary Table S2).
The instrumental variables for the proteins included in the analysis are listed in Supplementary Table S1. Detailed outcome of heterogeneity, pleiotropy, leave-one-out and single SNP analysis in the MR analysis are displayed in Supplementary Table S3-6.
[2.2. SMR ANALYSIS] line 123-124
The details of cis-eQTL data used in this study and the SMR results could be found in Supplementary Table S7.
[2.4. PHENOME-WIDE ASSOCIATION ANALYSIS (PheWAS):] line 154-155
The details of MR-PheWAS outcome can be found in Supplementary Table S8.
[2.5. PATHWAY AND FUNCTIONAL ANALYSIS:] line 167-169
As shown in Supplementary Table S9, the most significant pathways in the BP category were certain protein processes and the respiratory system.
[Supplementary Table S1-9]
|
Tables |
Title |
|
S1 |
The instrumental variants used in the MR analysis |
|
S2 |
MR results of the cis-pQTLs and RLS discoveries in IVW or Wald ratio methods |
|
S3 |
The detailed outcome of heterogeneity analysis in the MR analysis |
|
S4 |
The detailed outcome of pleiotropy analysis in the MR analysis |
|
S5 |
The detailed outcome of leave-one-out analysis in the MR analysis |
|
S6 |
The detailed outcome of singlesnp analysis in the MR analysis |
|
S7 |
The details of cis-eQTL used in the study and SMR results |
|
S8 |
The details of MR-PheWAS outcome |
|
S9 |
Detailed results from GO and KEGG enrichment analysis |
We sincerely thank you again for your constructive and valuable feedback. Your suggestions have significantly improved the quality and clarity of our manuscript, and we hope the revisions meet your expectations. We look forward to any further comments you may have.
Kind regards,
Ruiyi Qian

Reviewer 2 Report
Comments and Suggestions for Authors
Comments for Authors
The current article entitled “Identification of Causal Genes and Potential Drug Targets for Restless Legs Syndrome: a comprehensive Mendelian Randomization Study” has been reviewed. The article provides insights into potential drug targets and factors related to RLS through the integration of Mendelian randomization, gene co-localization, phenome-wide magnetic resonance imaging, gene enrichment, gene prediction, and molecular docking.
This work provides an important resource for researchers and clinicians in the field of RLS, as it lays a solid foundation for future research aimed at improving treatment. However, some modifications are needed for further improvement. Here are some;
The full form of the species should be given when the first time appears in both the abstract and in the remaining part of the manuscript.
Select appropriate keywords to attract researchers/readers. Increase visibility by using keywords obtained from international information networks.
The research gap should be clearly stated in the Introduction section.
Figure 1, legends should be improved, and a proper footnote should be given. All legends should have enough description for a reader to understand the table without having to refer back to the main text of the manuscript.
Table 1 and 2, the authors should properly mention the subscript and superscript, it should be checked throughout the manuscript, wherever applicable. For example, the necessary abbreviations should be given which are used in the present investigation.
Figure 2 is also lacking in depth and quality.
In the discussion section, researchers need to acknowledge sources of their information appropriately as indicated in the comment pane.
Some sentences and paragraphs are incomprehensible and therefore the text needs literary editing.
In the discussion section, it would be useful if you could add a concise summary of the main conclusions drawn from the study, as well as their implications.
The manuscript needs to be restructured and check for the typo errors (space missing).
Author Response
Dear Reviewer,
Thank you very much for your positive assessment of our work and for your constructive comments on our manuscript, “Identification of Causal Genes and Potential Drug Targets for Restless Legs Syndrome: a comprehensive Mendelian Randomization Study”. Your suggestions are invaluable in improving the quality and clarity of our paper. Below, we provide a point-by-point response to each of your recommendations.
Comment1: The full form of the species should be given when the first time appears in both the abstract and in the remaining part of the manuscript.
Response1: We appreciate your attention to detail. We have updated the manuscript to include the full form of species names.
[Abstract] line 24
Methods: All the analysis above were conducted in the context of Homo Sapiens.
[Introduction] line 82
In this study, we employed a comprehensive approach integrating Mendelian randomization, gene co-localization analysis, phenome-wide MR studies, gene enrichment analysis, gene prediction, and molecular docking to provide insights into potential drug targets and factors for RLS based on genetic evidence within the context of Homo Sapiens.
Comment2: Select appropriate keywords to attract researchers/readers. Increase visibility by using keywords obtained from international information networks?
Response2: Thank you for this suggestion. We have revised the keywords, using terms recommended according to MeSH-NCBI.
[Keywords] Restless Legs Syndrome; Drug Discovery; Mendelian Randomization; Proteomics.
Comment3: The research gap should be clearly stated in the Introduction section.
Response3: We have modified the Introduction section to explicitly state the research gap addressed by this study. Specifically, we added the content as follows:
[Introduction] line 55-63
Additionally, RLS prominently features a genetic underpinning according to a large-scale family study showing vertical transmission in 90% of all RLS family included[19, 20]. A large genome-wide association study (GWAS) identified 22 genetic loci associated with RLS and highlighted genetic correlations between RLS and neuropsychiatric traits[19]. Another recent large-scale meta-analysis by Schormair et al. expanded the number of known RLS risk loci to 164 and highlighted neurodevelopmental pathways and potential drug targets, with machine learning models showing high predictive accuracy for RLS risk[21]. However, the causal relationship between the targets and RLS has not been studied and discussed.
Comment4: Figure 1, legends should be improved, and a proper footnote should be given. All legends should have enough description for a reader to understand the table without having to refer back to the main text of the manuscript.
Response4: We have revised the legend for Figure 1 and added a detailed footnote to ensure it is self-explanatory. Additional description has been added to facilitate understanding without requiring reference to the main text.
[Figure 1] 102-114

Figure 1. (a) Forest plot displaying the positive findings from MR analysis. MAN1A2 exhibits a significant causal association (p_FDR < 0.05, p_Cochrane’s Q > 0.05 and p_Egger Intercept > 0.05) with RLS outcome. According to IVW, MAN1A2 was identified to mitigate the risk of RLS and exhibited a significant MR result (OR=0.68 [0.60,0.78], FDR =1.01E-04). (b) Volcano plots of the MR results between plasma proteins and RLS. The x-axis represents the log2 odds ratio (log2(OR)) of the causal effect of each protein, while the y-axis shows the negative logarithm of the p-value (-Lg(p-value)) for each association. The dashed horizontal line indicates the significance threshold (p = 0.05). The color gradient represents the level of significance, with warmer colors indicating stronger associations. The p-value of MAN1A2 is significantly lower than other proteins. All plasma proteins with p-value < 0.01, p_Cochrane’s Q > 0.05 and p_Egger Intercept > 0.05 were labeled, however, only MAN1A2 still showed significant association with RLS after FDR correction.
Comment5: Table 1 and 2, the authors should properly mention the subscript and superscript, it should be checked throughout the manuscript, wherever applicable. For example, the necessary abbreviations should be given which are used in the present investigation.
Response5: We have reviewed Tables 1 and 2 to correct and standardize the use of subscripts, superscripts, and abbreviations. All applicable abbreviations have been listed at the end of the paper.
[Table 1] line 125-128
The cis-eQTL data of MAN1A2 were extracted from only 11 of 52 tissues and were significantly expressed in the above five tissues. SMR, Summary-based Mendelian randomization; HEIDI, HEterogeneity In Dependent Instruments; *p_SMR < 0.05 and p_HEIDI > 0.05.
[Table 2] line 176-180
According to DSigDB database, 25 chemical compounds were predicted to be potentially effective intervention drugs to restless legs syndrome. Antimony potassium tartrate (CTD 00005424) had the lowest p-value, and sanguinarine (MCF7 DOWN) demonstrated the most favorable binding energy.
[List of Abbreviations] line 512-537
Abbreviation Full Term
BP Biological processes
CC Cellular components
CH-RLSq Cambridge-Hopkins Restless Legs Syndrome questionnaire
eQTL Expression quantitative trait locus
FDR False discovery rate
GO Gene Ontology
GWAS Genome-wide association study
HEIDI HEterogeneity In Dependent Instruments
IRLSSG International Restless Legs Syndrome Study Group
IVW Inverse variance weighted
KEGG Kyoto Encyclopedia of Genes and Genomes
LD Linkage disequilibrium
MF Molecular functions
MR Mendelian randomization
MR-PheWAS Phenome-wide MR analysis
NKA Na+-K+-ATPase
PP Posterior probabilities
pQTL Protein quantitative trait locus
RLS Restless legs syndrome
SMR Summary-based Mendelian randomization
SNPs Single nucleotide polymorphisms
STROBE-MR Strengthening the Reporting of Observational Studies in Epidemiology-Mendelian Randomization
UKB-PPP UK Biobank plasma proteomic project
Comment6: Figure 2 is also lacking in depth and quality.
Response6: Thank you very much for your suggestion on the quality of Figure 2. We have revised the figure to enhance clarity and detail, as well as adjusted the resolution to improve visual quality.
[Figure 2] line 156-163

Figure 2. Manhattan plot displaying MR-PheWAS findings. This plot illustrates the associations between MAN1A2 (from UKB-PPP) and 2407 phenotypes (from FinnGen) grouped into major biological categories. The x-axis represents phenotype categories, while the y-axis shows the negative logarithm of the p-values (-Lg(p-value)) for each association. The dashed horizontal line indicates the significance threshold (-log10(0.05)). MAN1A2 was not significantly related to the phenotypes after FDR correction. Four diseases with p-value < 0.0001 were labeled, but the relationship does not appear to be significant after FDR correction.
Comment7: In the discussion section, researchers need to acknowledge sources of their information appropriately as indicated in the comment pane.
Response7: We have reviewed the Discussion section and added appropriate citations where needed, as per your recommendation, to properly acknowledge all sources of information.
Comment8: Some sentences and paragraphs are incomprehensible and therefore the text needs literary editing.
Response8: We have conducted a thorough literary edit to improve readability, including rephrasing sentences and paragraphs to ensure clarity and coherence throughout the text.
Comment9: In the discussion section, it would be useful if you could add a concise summary of the main conclusions drawn from the study, as well as their implications.
Response9: In response to your suggestion, we have added a concise summary of the main conclusions and their implications in a seperate section to provide readers with a clear understanding of the study's key findings.
[5.Conclusion] line 463-470
This comprehensive Mendelian Randomization study provides robust evidence supporting MAN1A2 as a protective plasma protein for RLS, with various analyses enhancing the robustness. Drug prediction and molecular docking were implemented to validate the medicinal value of MAN1A2 as a therapeutic target. These insights not only deepen our understanding of RLS pathophysiology but also open new avenues for therapeutic intervention. Future studies are warranted to validate these findings and explore their clinical applications in managing and preventing RLS.
Comment10: The manuscript needs to be restructured and check for the typo errors (space missing).
Response10: We have carefully reviewed the manuscript for structural improvements and corrected any typographical errors, including missing spaces and minor formatting issues, to enhance readability and presentation.
We greatly appreciate your thoughtful review, which has contributed a lot to strengthening our manuscript. Thank you again for your time and valuable suggestions.
Kind regards,
Ruiyi Qian

Round 2
Reviewer 1 Report
Comments and Suggestions for Authors
I thank the authors for their efforts to reply to my questions. I am still not happy about exactly what data have been retrieved from the UK Biobank, or whether only summary UK biobank data have been used, from another research paper.
For example, in the response, the authors say that:
GWAS data for UKB-PPP are available through an interactive portal (https://metabolomips.org/ukbbpgwas/). UKB has catalogued the dataset in Category 1839, under ‘Field 30900’, described in greater detail online (https://biobank.ndph.ox.ac.uk/showcase/ label.cgi?id=1839).
But in the text the authors say that
GWAS 533 data for UKB-PPP are available through an interactive portal (h p://ukb-ppp.gwas.eu). UKB has 534 catalogued the dataset in Category 1839, under ‘Field 30900’, described in greater detail online 535 (h ps://biobank.ndph.ox.ac.uk/showcase/label.cgi?id=1839).
The links given are inconsistent and in the document do not appear to be formatted correctly.
Also under Methods, I would like to see a much more clear explanation of whether the authors have simply done a reanalysis of the summary data presented by Sun et al ("Plasma proteomic associations with genetics and health in the UK Biobank", Nature, 2023, doi: 10.1038/s41586-023-06592-6) rather than conducted a fresh primary analysis of their own.
In short, I would like the description and methods of the analysis of UK Biobank data to be substantially more clear, as the authors have NOT provided a UK Biobank application number, as described here:
https://www.ukbiobank.ac.uk/enable-your-research/apply-for-access
And the authors do NOT appear to be listed here
https://www.ukbiobank.ac.uk/enable-your-research/approved-research
Which makes me concerned that they are not analyzing UK Biobank data, they are just providing a secondary analysis of already-published data, and if so this must be made much more clear throughout the manuscript.
Author Response
Dear reviewer,
We really appreciate your thorough feedback regarding the clarity of our data source and methodology. We sincerely apologize for not adequately addressing this aspect in our original submission, leading to potential confusion.
To clarify, the data we utilized in our study are indeed derived from the published work of Sun et al. ("Plasma proteomic associations with genetics and health in the UK Biobank", Nature, 2023, doi: 10.1038/s41586-023-06592-6). We did not perform a primary analysis of UK Biobank data, but rather conducted a secondary analysis using the summary statistics provided in the aforementioned study.
In light of this feedback, we have carefully revised our manuscript to explicitly state the origin of the data and clarify that our work is a secondary analysis. We have updated the Methods section and Data Availability section to provide a detailed explanation of our approach, corrected any inconsistencies in the links cited, and ensured proper formatting throughout the text.
[4.1 DATA SOURCE] line 313-318
Proteomic data originates from the UK Biobank Pharma Proteomics Project (UKB-PPP)[24], wherein blood plasma samples were obtained from 54,219 UKB participants. Sun et al.[24] employed the antibody-based Olink Explore 3072 PEA to perform proteomic profiling, measuring 2,941 protein analytes and capturing 2,923 unique proteins. For detailed information on the sources of the summary data, please refer to the “Data Availability” section.
[Data Availability] line 501-503
We thank the UK Biobank, deCODE, Yang Lab, and FinnGen for sharing data and tools. The SMR tool was downloaded from “https://yanglab.westlake.edu.cn/software/smr”. GWAS data for RLS were downloaded from “https://www.decode.com/summarydata/”. We really appreciate Sun et al. for sharing UKB-PPP GWAS summary data on “https://metabolomips.org/ukbbpgwas/”. The raw UKB-PPP data can be obtained through an interactive portal ("http://ukb-ppp.gwas.eu"). UKB has catalogued the dataset in Category 1839, under ‘Field 30900’, described in greater detail online ("https://biobank.ndph.ox.ac.uk/showcase/label.cgi?id=1839"). GTEx v8 related data were downloaded from “https://www.gtexportal\”. Data from the 1000 Genomes Project was downloaded from “https://data.broadinstitute.org/alkesgroup”. The whole-blood eQTL data were downloaded from “https://eqtlgen.org”. The PheWAS phenotype data were obtained from “https://r10.finngen.fi”. All the data generated during this study are included in this article and its supplementary information files.
Additionally, we acknowledge the importance of citing the UK Biobank access and application requirements accurately. Since we are not conducting a direct analysis of raw UK Biobank data, but rather leveraging publicly available summary statistics, no UK Biobank application number is applicable in our case.
We are grateful for the opportunity to address these concerns, as it ensures the transparency and rigor of our work.
Best Regards,
Ruiyi Qian
Round 3
Reviewer 1 Report
Comments and Suggestions for Authors
Thank you for the additional clarifications. They are appreciated.